# T Cell Features in Glioblastoma May Guide Therapeutic Strategies to Overcome Microenvironment Immunosuppression

**DOI:** 10.3390/cancers16030603

**Published:** 2024-01-31

**Authors:** Agnese Losurdo, Antonio Di Muzio, Beatrice Claudia Cianciotti, Angelo Dipasquale, Pasquale Persico, Chiara Barigazzi, Beatrice Bono, Simona Feno, Federico Pessina, Armando Santoro, Matteo Simonelli

**Affiliations:** 1Medical Oncology and Hematology Unit, IRCCS Humanitas Research Hospital, Rozzano, 20089 Milan, Italy; agnese.losurdo@humanitas.it (A.L.); antonio.dimuzio@humanitas.it (A.D.M.); angelo.dipasquale@humanitas.it (A.D.); pasquale.persico@humanitas.it (P.P.); chiara.barigazzi@humanitas.it (C.B.); armando.santoro@humanitas.it (A.S.); 2Department of Biomedical Sciences, Humanitas University, Pieve Emanuele, 20072 Milan, Italy; federico.pessina@hunimed.eu; 3Laboratory of Translational Immunology, IRCCS Humanitas Research Hospital, Rozzano, 20089 Milan, Italy; beatrice.cianciotti@humanitasresearch.it (B.C.C.); simona.feno@humanitasresearch.it (S.F.); 4Department of Neurosurgery, IRCCS Humanitas Research Hospital, Rozzano, 20089 Milan, Italy; beatrice.bono@humanitas.it

**Keywords:** glioblastoma, T cells, tumor microenvironment, immune check-point blockade, vaccinations, adoptive cell therapy

## Abstract

**Simple Summary:**

Glioblastoma is a lethal primary brain tumor, and so far, immunotherapeutic strategies have not significantly improved patients’ prognosis, both in newly diagnosed and recurrent settings. Understanding the features of the immune environment of the central nervous system is crucial to designing new treatment strategies able to counteract site-specific immunosuppressive and pro-tumorigenic factors. Here, we review next-generation immunomodulating therapeutic strategies such as immune check-point blockade, vaccinations, and adoptive cell therapies, aiming to re-shape the tumor microenvironment and restore active anti-tumor immunity. The door for immunotherapeutic strategies in glioblastoma treatment is not completely closed; researchers should test combinatorial treatments and design trials with solid translational analyses to gain more close and deep insight into on-treatment modifications of the tumor microenvironment.

**Abstract:**

Glioblastoma (GBM) is the most aggressive and lethal primary brain tumor, bearing a survival estimate below 10% at five years, despite standard chemoradiation treatment. At recurrence, systemic treatment options are limited and the standard of care is not well defined, with inclusion in clinical trials being highly encouraged. So far, the use of immunotherapeutic strategies in GBM has not proved to significantly improve patients’ prognosis in the treatment of newly diagnosed GBM, nor in the recurrent setting. Probably this has to do with the unique immune environment of the central nervous system, which harbors several immunosuppressive/pro-tumorigenic factors, both soluble (e.g., TGF-β, IL-10, STAT3, prostaglandin E2, and VEGF) and cellular (e.g., Tregs, M2 phenotype TAMs, and MDSC). Here we review the immune composition of the GBMs microenvironment, specifically focusing on the phenotype and function of the T cell compartment. Moreover, we give hints on the therapeutic strategies, such as immune checkpoint blockade, vaccinations, and adoptive cell therapy, that, interacting with tumor-infiltrating lymphocytes, might both target in different ways the tumor microenvironment and potentiate the activity of standard therapies. The path to be followed in advancing clinical research on immunotherapy for GBM treatment relies on a twofold strategy: testing combinatorial treatments, aiming to restore active immune anti-tumor responses, tackling immunosuppression, and additionally, designing more phase 0 and window opportunity trials with solid translational analyses to gain deeper insight into the on-treatment shaping of the GBM microenvironment.

## 1. Introduction

Adult-type diffuse gliomas are the most common primary brain tumors, accounting for around 70% of all malignant central nervous system (CNS) malignancies [1]. They are characterized by a very poor prognosis, with the most prevalent and aggressive form, glioblastoma (GBM), bearing a survival estimate below 10% at five years [1]. The WHO classification has evolved in recent years, with the most recent fifth edition, published in 2021, introducing major changes to both tumor nomenclature and grading and integrating molecular parameters into established histology and immunohistochemistry [2]. On the basis of Isocitrate dehydrogenase (IDH) mutations and 1p/19q co-deletion, adult-type diffuse gliomas are now subdivided into three major subgroups: IDH-mutant (IDH-mut) astrocytoma, IDH-mut and 1p/19q-codeleted oligodendroglioma, and IDH-wildtype (IDH-wt) GBM [2].

The standard treatment approach for newly diagnosed GBM (nGBM) comprises a multimodality strategy consisting of maximally safe surgical resection followed by radiotherapy (RT) with concurrent temozolomide (TMZ) chemotherapy and subsequent TMZ maintenance for 6 months [3]. Despite this aggressive multimodal treatment program, virtually all GBM patients experience disease recurrence. The different treatment modalities used for tumor progression or recurrence are typically individualized, taking into account different factors, such as the radiological pattern of relapse, time since diagnosis, previous treatment, and, above all, patients’ performance status and neurological function. On these grounds, surgery, RT, or medical therapy might be proposed. Unfortunately, systemic treatment options at recurrence are limited, and the standard of care at this stage is not well defined, with inclusion in a clinical trial being highly encouraged.

The use and testing in clinical trials of immunotherapeutic strategies in GBM have to deal with the unique immune environment of the CNS. Indeed, the CNS has been long considered an immune-privileged organ due to the presence of the blood-brain barrier (BBB) and the lack of a dedicated lymphatic system [4,5,6]. However, more recent evidence has fully demonstrated the presence of lymphatic vessels within the dural sinuses, allowing the outflow of specific brain antigens and the inflow of leukocytes through cerebrospinal fluid and cervical lymphatics. [7]. Another critical barrier to effective antitumor immunity has been traditionally represented by the wide array of complementary and overlapping immunosuppressive mechanisms of diffuse gliomas, acting both in the tumor microenvironment (TME) and systemically. Systemic immunosuppression, especially in the context of cell-mediated immunity, has long been described in glioma patients, who additionally often require corticosteroids to attenuate tumor-associated cerebral edema and receive cytotoxic chemotherapy, which may trigger prolonged lymphopenia. Many cellular and soluble immunosuppressive factors may be described in the context of glioma microenvironment, including, but not limited to (i) down-regulation of the major histocompatibility complex (MHC), masking glioma cells from immune attacks; (ii) overexpression of a wide range of potent soluble immunosuppressive cytokines, including transforming growth factor (TGF)-β, interleukin (IL)-10, signal transducer and activator of transcription 3 (STAT3), prostaglandin E2, and vascular endothelial growth factor; (iii) recruitment of immunosuppressive cell types such as regulatory T-cells (Tregs), tumor-associated macrophages (TAMs) polarized towards the tumorigenic M2 phenotype, and myeloid-derived suppressor cells (MDSC); (iv) activation of intracellular pathways involved in T cell immune tolerance and suppression of T cell cytotoxic activity (e.g., IDO, FASL, PD-L1, STAT3) [4,8,9]. In addition, GBM cells are capable of inducing sequestration of T cells into the bone marrow through downregulation of sphingosine 1-phosphate receptor 1 (S1P1) in T cells [10].

Here we review the immune composition of the GBMs TME, specifically focusing on the phenotype and function of the T cell compartment. Moreover, we give hints on the therapeutic strategies that, by interacting with TILs and shaping the TME in different ways, might both potentiate the activity of standard chemotherapy and/or RT and help overcome resistance to these treatments.

A total of around 130 articles, of which around 50 present clinical trial results, published between 2000 and 2023 worldwide, were selected through searching PubMed, ClinicalTrials.gov, Scopus, and the Cochrane Library. Keywords such as Glioblastoma, Central Nervous System, Immunotherapy, Vaccine, CAR-T, Check Point Inhibitors, Tumor Microenvironment, TILs, immune cell subpopulations, and their combinations were used in the search. The titles and abstracts were scanned to be appropriate to be included in this study. Furthermore, the list of references from the selected reviews and articles was searched for more references.

## 2. Gliomas Immune Landscape

### 2.1. Microglia and Macrophages

Microglia and macrophages, together accounting for the majority (around 30%) of TME cells, constitute the innate immunity of the CNS and are responsible for the maintenance of normal brain function and homeostasis [5]. Tissue-resident microglia develop from embryonic yolk sac progenitor cells and are not replenished postnatally from peripheral blood, while bone-marrow-derived macrophages (BMDMs) are generated by circulating monocytes recruited to the brain parenchyma in pathological conditions and may be replenished from peripheral blood, especially in cases of disruption of the BBB (e.g., irradiation, chemicals, tumor progression) [6,11,12]. Whether microglia and BMDMs have distinct roles and functions, and how to clearly distinguish them using cell surface markers in flow cytometry analysis, is still a topic of investigation. Joyce’s group proposed the cell surface marker integrin alpha 4, ITGA4/CD49D, to discriminate CD49D^low^ microglia from CD49D^high^ BMDMs [5,12]. Furthermore, the proportion of these two populations seems to differ based on tumor grade and IDH mutational status: microglia predominate in low-grade IDH-mut gliomas, whereas BMDMs are more prevalent in high-grade IDH-wt gliomas [5,13]. In terms of spatial localization, BMDMs are more frequently localized within the GBM parenchyma, while microglia tend to reside at the tumor periphery [14]. Both tissue-resident macrophages and BMDMs are able to give rise to TAMs, highly plastic cells that play a pivotal role in connecting tumor cells with adaptive immunity in the TME, regulating angiogenesis and epithelial-mesenchymal transformation [15]. Based on their markers’ expression and cytokine secretion profile, TAMs may be polarized into M1- or M2-phenotypes, exerting pro-inflammatory/anti-tumorigenic and immune-suppressive/pro-tumorigenic features, respectively. The M1 phenotype is characterized by CD80/CD86 surface expression and IL-12/IL-23/TNFα/IL-6 secretion, while the M2 phenotype is present with CD163/CD204/CD206/ARG1 surface markers and IL-10/TGFβ production [16,17]. Nevertheless, the M1-M2 phenotypic continuum has long been disputed because TAMs in GBM tend to be pro-tumorigenic and lack expression of key molecules involved in T cell co-stimulation (e.g., CD40, CD80, and CD86). Thus, various studies have implicated TAMs, via secreted pro-tumoral factors (such as TGFβ, IL-6, IL-1β, IL-4, IL-10, and metallopeptidases), in promoting tumor cell proliferation, epithelial-mesenchymal transformation, and angiogenesis, along with the induction of an immunosuppressive environment by attracting Tregs and MDSC [18,19].

### 2.2. Neutrophils

Neutrophils are the most abundant leukocyte population in the blood stream, and mechanisms of recruitment at the tumor site by GBM neoplastic cells have not been totally elucidated so far. Possibly, soluble factors, such as IL-6 and IL-8, produced upon activation of the Fas/Fas-L interaction, are responsible for circulating neutrophils and their infiltration into the tumor [20,21]. At the tumor site, similarly to macrophages, tumor-associated neutrophils (TAN) possess two main polarized statuses that can switch into one another and play opposite roles on tumor progression: the so-called N1 with antitumoral phenotype induced mainly by IFN-β stimulation and the tumor-promoting phenotype N2 elicited by TGF-β, G-CSF, and IL-6 [22]. Globally, clinical data suggest that TAN have a negative prognostic value in GBM, and accordingly, CD11b+ neutrophils have been proven to be enriched in IDH-wt compared to IDH-mut and in recurrent compared to newly diagnosed GBMs [5]. Soluble factors released by pro-tumorigenic polarized TANs, such as S100A4, have been described to be active in pathways supporting angiogenesis and inducing the transformation of glioma stem cells towards a mesenchymal phenotype, thus proving the role of TANs in indirectly promoting tumor cell proliferation and invasion [23]. Moreover, TANs support tumor necrosis, which is not only a diagnostic hallmark for GBM (induced by ischemia) but also correlates with tumor aggressiveness and poor outcomes. Concerning spatial location, GBM-infiltrating PD-L1+ TANs were found to be in close proximity to PD-1-expressing CD8+ T cells, corroborating their potential role in mediating immunosuppression [24]. In the circulation, a high neutrophil count and an elevated neutrophil-to-lymphocyte ratio (NLR) correlate to poor prognosis (both when measured before surgery and after RT plus TMZ concurrent therapy), IDH-wt status, and a positive response to anti-VEGF-A therapy (bevacizumab) [25,26,27,28,29]. In contrast, intratumoral neutrophilia was previously associated with the development of drug resistance to bevacizumab [23]. Potentially, this contrasting predictive effect is linked to the secretion by GBM cells of Granulocyte-Colony Stimulating Factor (G-CSF), a growth factor for neutrophils that is also responsible for a specific type of angiogenesis that correlates to bevacizumab overall survival (OS) benefit [29].

### 2.3. Dendritic Cells

Dendritic cells (DCs) are myeloid-derived, professional antigen-presenting cells (APCs), with the crucial role of presenting antigen captured in peripheral tissues to T lymphocytes present in secondary lymphoid organs in order to elicit immune responses [30]. Historically, DCs were thought to be present in all human tissues except for the brain [31], nevertheless, in more recent years, DCs have been clearly identified, even though in a very small number, also in the CNS and infiltrating both primary brain tumors and brain metastasis TME, residing at perivascular and intraparenchymal inflammatory sites [5,32]. DCs arise from lympho-myeloid hematopoiesis in the bone marrow and may be distinguished into at least two major subsets: conventional/myeloid DCs (cDCs) and plasmocytoid DCs (pDCs) [33]. Specifically, pDCs harbor the peculiar ability to quickly secrete large amounts of type I interferons (IFN I) in response to pathogens when they also undergo important phenotypic changes, such as the acquisition of a dendritic morphology (pDCs appear round in shape before activation) and the upregulation of MHC, which enable pDCs to engage and activate naïve T cells [34]. In tumors, pDCs may play different and sometimes contrasting roles on tumor progression and immune evasion: they can promote tolerance by presenting antigen to CD4+ T cells and inhibiting their activation or inducing Treg, but they also produce large amounts of IFN-α and prime CD4+ and CD8+ T cells, inducing an effective antitumor response [35,36,37,38]. Even if DCs are present in a very small number in GBM, where notably the most important APC is microglia, growing clinical interest in the design of DC vaccines with therapeutic intent has risen interest in this neglected immune cell subtype.

### 2.4. T Cells

#### 2.4.1. Conventional T Cells

CD4+ and CD8+ T cells account for approximately 5% of the total leucocytes (CD45+ cells) infiltrating the malignant gliomas and represent the most abundant adaptive immune cells in the glioma TME [39]. T cell infiltration is more abundant in IDH-wt compared to IDH-mut GBM [5,40] (Figure 1). Similarly to other tumors, the T-cell-mediated immune response in malignant gliomas is dampened by T cell exhaustion. T cell exhaustion is a hyporesponsive T cell state occurring as a consequence of chronic antigen exposure. Exhausted cells are characterized by decreased proliferative ability, reduced effector functions, and upregulation of multiple inhibitory receptors [41]. Originally described in chronic infections, T cell exhaustion is a physiological mechanism aimed at reducing excessive tissue damage during inflammation. One of the strategies to hijack T cell exhaustion is the use of immune checkpoint blockade (ICB) to block the interactions among immune receptors and their cognate ligands expressed by tumor cells and APC in TME. This strategy reinvigorates the anti-tumor T cell response and has proved successful in clinical practice for some tumors, but has not proved effective so far for GBM [42]. A deeper understanding of T-cell exhaustion mechanisms in GBM is needed to elucidate the causes of ICB failure and to build effective immunotherapy for GBM patients. Flow cytometry analysis of 21 GBM samples and matched peripheral blood identified LAG-3, TIM3, TIGIT, and CD39 as possible immunotherapeutic targets beyond PD-1 and CTLA-4; T cells co-expressing PD-1, LAG-3, TIGIT, and CD39 proved unable to produce IFNγ, IL-2, or TNFα [43]. In a similar study, Davidson and colleagues characterized TILs from GBM patients in three independent cohorts [44]. Using flow cytometry and mass cytometry, they found a higher frequency of PD-1+ T cells in the tumor compared to matched peripheral blood or healthy donors [44]. Expanded PD-1+ T cells co-expressed TIM-3, LAG-3, CTLA-4, and CD39. Contrary to the previous TIL characterization [43], sorted PD-1+ TILs showed higher IFNγ production upon TCR triggering compared to sorted PD-1- TILs [44]. In a recent paper from the group of Dr. Joyce, the TME of both patients with primary (GBM) or metastatic brain tumors was interrogated at the single-cell level; the analysis revealed similarities and differences in T cell phenotype and function, with the most pronounced differences observed in the relative abundance of CD39+ potentially tumor-reactive T cells [40]. In GBM, CD39+ T cells were very poorly represented in an immune microenvironment background displaying low levels of inflammatory and T cell-recruiting cytokines [40]; indeed, the expression of CD39 on the CD8+ TILs has been correlated to the presence of active, tissue-resident, tumor-specific T cells, frequently correlating to different tumor types, with a better prognosis [45,46].

Additional immune suppressive mechanisms beyond T cell exhaustion have been recently described in the GBM microenvironment. A single-cell RNA-seq analysis of GBM samples identified S100A4 as a highly expressed gene in infiltrating T cells; its expression was associated with a poor prognosis in GBM patients. S100A4-/- mice showed increased OS and increased T cell infiltration compared to wild-type hosts when injected with primary glioma cells. In addition, T cells infiltrating S100a4-/- glioma-bearing mice displayed high proliferation and IFNγ production [47]. S100A4 is a calcium-binding protein, and its role in different biological processes (e.g., angiogenesis, invasion, stemness) has been described in different tumor cell types [48]. However, its role in T cell response needs to be deeply addressed. Using the same in silico approach in another scRNA-seq dataset from 8 GBM patients, Ravi and colleagues identified IL-10-producing HMOX1+ IBA1+ myeloid cells as main interactors with exhausted T cells [49]. In a human neocortical GBM model engrafted with autologous patient-derived T cells, chemical depletion of myeloid cells reduced IL-10 production by myeloid cells and increased IL-2 and IFNγ production by T cells; in addition, pharmacological IL-10 scavenging increased the frequency of effector GZMB+ T cells [49].

Immunosenescence is also considered a hypofunctional T cell status, and it is characterized by a loss of replicative capacity and telomere shortening. Senescent T cells express CD57 and lack CD28 as surface markers. The analysis of 42 human cytomegalovirus-positive (HCMV-positive) GBM patients revealed that CD57+CD28−CD4 T cells are expanded and associated with poor prognosis [50], thus pointing out immunosenescence as a relevant pathway to boost T cell response in GBM.

#### 2.4.2. T Regulatory Cells

The T-cell response in GBM is also dampened by the presence of immunosuppressive CD4+CD25+FoxP3+ Tregs. Tregs are increased in GBM patients and suppress CD8+ T cell proliferation and effector functions [51,52]. Similarly, the frequency of a subset of GITR+ Treg is increased in the CD4 compartment in a GBM mouse model [52]. Treatment with an anti-CD25 neutralizing antibody reduced their suppressive capacity, thereby increasing CD8+ T cell cytotoxic functions; interestingly, a combination of anti-CD25 treatment and DC vaccination resulted in tumor rejection in 100% of challenged mice [52]. In the same line of evidence, an increased fraction of GZMB+ and GITR+ Tregs has been described in the same GBM murine model. Intracranial, but not systemic, administration of anti-GITR antibodies induced Treg depletion and increased the OS of treated mice compared to controls [53].

While the role of Tregs in the maintenance of an immunosuppressive microenvironment in GBM is well established, the mechanisms of their recruitment are less elucidated. Tumor-derived Indoleamine-2,3-dioxygenase (IDO) accumulation has been proposed as a mechanism of Tregs accumulation in GBM. A high IDO level is associated with a reduced OS in GBM patients. IDO-deficient glioma mouse models showed an increased survival advantage compared to IDO-competent mice; this advantage was associated with a low frequency of brain-infiltrating Tregs [54].

All this evidence pointed out relevant immunotherapeutic targets in glioma TME, whose targeting might be beneficial in restoring the anti-tumoral response in glioma.

### 2.5. Tumor Microenvironment: Specific Changes in Recurrent GBM

A specific mention is due to changes occurring in the microenvironment of rGBM, which is known to harbor major aggressiveness pathways due to therapeutic pressure and clonal selection [55]. In a window-of-opportunity clinical trial of pembrolizumab given before and after surgery for rGBM, the TME revealed very poor T cell infiltrate, with CD68+ macrophage preponderance [56]. In spite of the small sample size (12 patients) and the previous exposure to ICB, which might alter the phenotype and function of infiltrating immune cells, this study revealed a predominant cluster of CD68+HLA-DR+CD56+B7H3Low TAMs by CyTOF analysis in patients treated with pembrolizumab and a significant increase in GZMB immunohistochemical expression in pembrolizumab-treated patients compared to newly diagnosed GBMs [56]. A significant increase in tumor infiltrating myeloid populations, namely CD68+ macrophages and CD11b+ cells, both in the tumor bulk and in the infiltrative regions, was also observed in autopsy brain specimens of rGBM patients treated with anti-angiogenic therapy; of note, an increased number of tumor infiltrating CD11b+ cells proved to correlate with decreased OS in these patients [57]. A recent study, analyzing paired primary-recurrent IDH-wt GBM samples using RNA-sequencing, showed preferential mesenchymal progression and enrichment in TAM signatures in rGBM without changes in hallmark GBM-associated genes [58]. The ontogeny of TAMs in rGBM has been investigated in a study analyzing both human and mouse GBM samples at the single-cell level: microglia-derived TAMs were predominant in newly diagnosed tumors, while BMDMs were more abundant in rGBM samples, especially in hypoxic TME [59]. Moreover, treatment-specific TME alterations are reported in the literature, highlighting the crucial importance of in-depth characterization of the unique TME of rGBM, which exhibits diverse plastic modifications when responding to environment-specific stimuli. For example, in a recent in vivo study, the TME of GBM undergoing RT plus concurrent ICB was found to be enriched in CD103+ Tregs, displaying elevated cholesterol and lipid metabolism pathways and a highly suppressive transcriptional program [60].

## 3. Immunotherapy

### 3.1. Targeting the PD-1/PD-L1 Axis, Alone or in Combination with Anti-CTLA4

The striking results of immunotherapy with checkpoint blockade in different cancer types have fueled numerous clinical studies in the field of neuro-oncology. After evidence in preclinical mouse models of TIL infiltration of brain tumors and truly promising data of tumor regression mediated by inhibition of specific co-stimulatory receptors (e.g., CTLA-4 and PD-1/PDL-1) [8,61,62], clinical trials testing checkpoint blockades in GBMs were designed. Different putative targets of immunotherapeutic strategies in GBM are depicted in Figure 2a. Table 1 summarizes the clinical trials with published results exploring the use of anti-PD-1/PD-L1 therapy in GBM in different phases of clinical development. The results of the early-phase part of the CheckMate 143 trial, which tested different ICB strategies in rGBM [42], led to the opening of the first randomized phase III study exploring the efficacy of anti-PD-1 agent nivolumab in rGBM [63]. A total of 439 patients with rGBM who progressed after the traditional Stupp regimen were randomized 1:1 to receive nivolumab or the anti-VEGF mAb bevacizumab, with OS as the primary end-point [63]. This study failed to demonstrate a statistically significant difference between treatment groups in terms of both median OS (mOS) and median progression-free survival (mPFS) (mOS: 9.8 and 10 months for nivolumab and bevacizumab, respectively; mPFS: 1.5 and 3.5 months for nivolumab and bevacizumab, respectively) [63]. Post hoc hypothesis-generating analyses conducted in previously pre-specified patient subgroups showed that patients with methylated MGMT promoter and without steroid use at baseline (*n* = 31) treated with nivolumab reached the longest mOS of 17 months [63]. The role of ICB in nGBM was further explored in the adjuvant setting by two different phase III trials: the CheckMate 498 trial randomized newly diagnosed GBM (nGBM) patients with unmethylated MGMT promoter to receive either nivolumab plus RT or standard TMZ plus RT, while the CheckMate 548 trial randomized methylated MGMT promoter patients to receive nivulomab plus TMZ and RT or placebo plus TMZ and RT [64,65]. Final results of CheckMate 498 showed an overlapping OS in the two groups (mOS 13.4 and 14.9 months for the experimental and control arm, respectively), with a similar incidence of grade 3/4 TRAEs (21.9% and 25.1% in the experimental and control arm, respectively) [64]. These findings may lead to a twofold speculative conclusion: on the one hand, knowing the poor efficacy of TMZ in unmethylated MGMT promoter patients [66], nivolumab may seem detrimental in this population; on the other hand, we might conclude that TMZ continues to provide an albeit limited benefit in this population. Similarly, the CheckMate 548 trial did not meet its primary end-points, with no statistically significant differences in terms of both OS and PFS when nivolumab was added to standard TMZ and RT (mOS: 28.9 vs. 32.1 months and mPFS: 10.6 vs. 10.3 months in the experimental and placebo arms, respectively), even in the subgroup of patients without baseline corticosteroids [65]. Taking together the data that emerged from these phase III trials evaluating a global population of 1715 patients, of whom 496 were in the relapsed setting and the remaining at first diagnosis, it seems reasonable to conclude that the use of anti-PD-1 agents is not effective as a treatment for GBM in different phases of the disease course. However, as highlighted by phase 0 and translational trials, the timing of ICB may be crucial in influencing the quality, strength, and durability of the immune response. In the seminal paper by Cloughesy and colleagues, testing pembrolizumab in the neo\adjuvant setting in rGBM, patients randomized to receive immunotherapy in the neoadjuvant setting had a significantly prolonged survival compared to those in the adjuvant arm [67]. The significance of these outcomes were strengthened by translational data showing the upregulation of T cell and IFN-γ-related gene signatures and the enhanced clonal expansion of T cells (with focal induction of PD-L1) in the TME, underlying the potential capacity of ICB to stimulate/boost anti-tumor T cell responses and generate specific immunologic memory when given to an intact (not pre-treated) TME [67]. Despite the poor survival results coming from randomized clinical trials conducted so far, the chapter on immunotherapy with ICB in GBM may not yet be concluded. Several early-phase trials involving combinations of ICB, anti-angiogenic agents, or integration with other innovative immunotherapeutic strategies such as cellular or peptide vaccines are under intensive clinical investigation, with the aim of reverting immunoresistance, re-sensitizing GBM tumor cells, and finally avoiding immunotolerance (Table 2).

### 3.2. Other Co-Stimulatory and Co-Inhibitory Pathways

PD-1/TIGIT-axis emerged as putative immunotherapy targets in GBM in the analysis of TCGA datasets from 153 GBM patients. GBM patients with upregulated PD-1 and TIGIT genes and their cognate ligand genes displayed reduced OS and PFS compared to patients who did not upregulate the same genes [78]. In a GBM mouse model, treatment with anti-TIGIT/anti-PD-1 resulted in increased survival of the mice, reduced tumor volume, and increased immune infiltration in tumor lesions compared to single ICBs or isotype treatments [78]. A phase I clinical trial has been set up to study an anti-TIGIT antibody (AB154) in combination with anti-PD-1 therapy in rGBM (NCT04656535). In similar preclinical GBM models, dual blockade of TIM-3 and PD-1 or dual blockade of TIM-3 and BTLA [79] resulted in increased OS in treated mice compared to controls or single ICB treatment. Interestingly, both of these combination therapies proved more effective in increasing the proportion of cytokine-producing CD8+ T cells and in reducing the frequency of immunosuppressive Tregs within the tumor compared to single ICB treatment [79,80]. A phase I trial of anti-TIM3 in combination with anti-PD-1 treatment is now addressing the efficacy of this combination strategy in patients with rGBM (NCT03961971). Another marker of early T cell exhaustion, LAG3 (CD223), whose inhibition has been shown in preclinical models to be able to skew CD4+ cells away from the Treg phenotype [81], has been tested in a phase I trial for rGBM, alone or in combination with anti-PD-1 therapy (NCT02658981). Preliminary results of the first 44 patients enrolled indicated a mOS of 8 months for the anti-LAG3 monotherapy (with no dose-limiting toxicities (DLTs) for the highest safe dose of 800 mg) and 7 months for the combination therapy with anti-PD-1 [82]. Preclinical evidence demonstrated that another strategy for reshaping the TME, promoting effector T cell function, and inhibiting Tregs might be the activation of the glucocorticoid-induced TNFR-related gene (GITR) [83]. Nevertheless, GITR activating monotherapy has not proved effective in solid tumors [84], and combination strategies, together with another ICB (e.g., anti-PD-1) [85], chemotherapy [86], or RT [87], have been tested in pre-clinical and clinical trials to improve GITR agonism. An initial signal of the potential efficacy of these combination strategies came from a phase II trial for rGBM, where patients were treated with an anti-PD-1 plus a GITR agonist (INCAGN01876) together with fractionated stereotactic RT (FSRT) at disease recurrence (Cohort A) or, if amenable to surgery, patients were allocated to receive an anti-PD-1 plus INCAGN01876, with (Cohort B1) or without (Cohort B2) pre-surgical FSRT (NCT04225039). The combination of an anti-PD-1 plus INCAGN01876 and FSRT without surgical resection did not demonstrate efficacy in patients with rGBM, but interestingly, patients in Cohort B1 had significantly longer mPFS and mOS compared to patients in Cohort B2 (PFS 11.7 vs. 2.0 months, respectively; *p* = 0.0002, and mOS 20.1 vs. 9.4 months, respectively; *p* = 0.001), with a concomitant increase in inflammatory cytokine responses and proliferative T cells [88].

Lastly, scRNA-seq analysis of 31 GBM samples identified the NK-marker CD161 (KLRB1) as a novel potential immunotherapeutic target; specifically, the authors identified CD161 as an overexpressed gene in cytotoxic CD8 T cells [89]. CRISPR/Cas9-mediated CD161 disruption in T cells redirected against the NY-ESO-1 tumor antigen increased T-cell-mediated glioma cell killing in vitro and their anti-tumor efficacy in vivo [89].

### 3.3. Vaccines

Vaccines are among the therapies that capitalize on the established efficacy of the immune system in fighting cancer; they work by activating the adaptive immune system, particularly T cells, against antigens that might be set up and injected from outside to be carried on the membrane of APCs, or activated APCs can be administered directly [90]. The optimal source of antigens to be used for APC loading is still controversial, as the “perfect” antigen should bear tumor specificity in order to avoid autoimmune attacks on healthy cells but also be broadly present at the tumor site, allowing targeting of as many cancer cell clones as possible. In view of this, we can distinguish so-called tumor-associated antigens (TAAs), ubiquitous non-mutated proteins that are upregulated in tumors compared to normal tissues (e.g., EGFR, IL-13Rα2, and gp100), and tumor-specific antigens (TSAs), produced and expressed only by tumor cells (e.g., EGFRvIII). For example, taking into account GBM therapy, a phase III trial evaluating Rindopepimut, a peptide vaccine targeting the EGFR mutation EGFRvIII, did not show any survival benefit in patients with nGBM [91]. In contrast, favorable results were reported for a peptide vaccine against the surviving molecule, a TAA, namely SurVaxM, tested in a phase II trial in resected nGBM, showing a mPFS of 11.4 months and a mOS of 25.9 months [92].

Here, we will focus on so-called DC-vaccines, specifically describing the results of the most recent phase II and III randomized clinical studies. As previously shown, DCs might be distinguished into conventional/myeloid DC 1-2 (cDC1, cDC2) and plasmacytoid DC (pDC); the cDC1 subtype appears to have a higher affinity for activating CD8+ T cells, while cDC2 is for CD4+ T cells [93]. In manufacturing DC-based vaccines, it is unknown which subsets or combinations are most efficient; nevertheless, the most common technology involves the production of monocyte-derived DC ex vivo by in vitro stimulation of monocytes isolated from peripheral blood mononuclear cells (PBMCs). The specific methodology to isolate and culture/differentiate monocytes into cDCs is still controversial, with different cell culture media, serum sources, and required concentrations of GM-CSF and IL-4 used in different studies [94]. Also, the sources of antigens used to activate autologous ex vivo-generated DC may be different in nature, such as whole-tumor lysate, antigenic peptides, viral transfection, or messenger RNA (mRNA) electroporation. A schematic representation of the DC vaccine strategies is depicted in Figure 2b. The development of trials with DC vaccines in GBM has been hampered by issues common to other solid tumors, such as the selection of the best antigen to stimulate an effective and sustained immune response, the timing of administration (e.g., neo-adjuvant setting or advanced disease), the expansive cost of manufacturing, and the long turnaround time. While specific problems in developing DC vaccines for GBM include the low sample size given the disease’s low incidence, severely limiting the ability to conduct large trials with control arms, and the short disease median OS, especially for relapsed GBM, greatly reducing the time window available to produce and demonstrate the efficacy of therapies aiming to activate the immune system [95]. A list of trials testing DC vaccination in GBM with reported results is displayed in Table 3. The phase III trial assessing tumor lysate-loaded DC vaccine (DCVax-L) in patients with nGBM, although formally meeting its primary endpoint, with a mOS of 19.3 versus 16.5 months (HR = 0.80; 98% CI, 0.00–0.94; *p* = 0.002) for patients receiving DCVax-L and placebo plus TMZ, respectively, was very controversially conducted [96]. Important limitations, both in the nature and methodology of this study, need to be highlighted. Firstly, in the original trial design, the primary end-point was PFS determined by MRI, but due to difficulties in distinguishing progressive disease from pseudo-progression related to immune-related necrosis or inflammation, OS was used for the definitive statistical analyses as the primary end-point. Moreover, this study was presented as randomized, double-blind, with a crossover design, but due to the high number of patients accessing the crossover phase, the placebo group was depleted, and the survival analyses were conducted using a post hoc external control population for whom individual patient level data were not available, and thus a propensity score analysis was not feasible. Thus, taking all this into account, the trial should not be considered to answer the question on DCVax-L activity in GBM, and its post hoc conclusive analyses are to be accounted for only as hypothesis-generating [97].

As GBM cells might contain sequences and express the viral gene products of CMV [102], investigators at Duke University designed a set of clinical trials assessing a vaccination with CMV pp65-specific autologous DCs. The ATTAC trial (testing the pre-conditioning of the vaccine site with tetanus/diphtheria (Td) toxoid) [NCT00639639], the ATTAC-GM (expanded cohort receiving GM-CSF containing autologous CMV DC vaccines with dose-intensified TMZ) [NCT00639639], and the ELEVATE trial [NCT02366728] [103]. The vaccination was administered with TZM in nGBM after surgical resection and chemoradiation and proved very encouraging 5-year OS rates of 33.3% (mOS 38.3 months; 95% CI, 17.5–∞) and 36.4% (mOS 37.7 months; 95% CI, 18.2–109.1) for ATTAC-Td and ATTAC-GM, respectively, without significant TRAEs [103]. In the confirmatory double-blind randomized phase II trial (ELEVATE), the 3-year OS for the Td arm was 34% (95% CI, 19–63%) compared to 6% (95% CI, 1–42%) in the unpulsed DCs arm [104]. Translational analyses in the ATTAC study showed a steady increase in the Treg population after the first dose of maintenance TMZ, followed by an increase in peripheral CD8+ T cells and the CD8+:Treg ratio after pp65 vaccination [105]. Therefore, in the ELEVATE trial, the investigators added a second experimental arm using the anti-Treg monoclonal antibody Basiliximab (anti-IL-2rα) in order to test whether inhibition of the initial increase in the Treg pool may serve as a boost for the T-mediated response induced by the pp65 DC vaccine. Preliminary data did not show a survival advantage in the Basiliximab arm compared with patients treated with DC vaccine plus Td pre-conditioning, with a mOS of 19 (16.7–25.6) and 20 (10.26–NA) months, respectively [NCT02366728].

While promising, therapies in GBM based on DC vaccines are far from clinical practice, and we still await the results of larger, more advanced-phase trials to demonstrate a substantial clinical benefit. Translational analyses will be crucial to understanding how to develop a vaccine able to induce strong and long-lasting T-cell-mediated immune activation against GBM cells. Many early-phase clinical trials are attempting to increase the immunogenic potential of the DC vaccines in GBM; specifically, the most promising ongoing strategies are testing the combination of DC vaccines with ICB or local therapies such as RT or medicated wafers [101,106].

### 3.4. Adoptive Cell Therapy and Chimeric Antigen Receptor T Cell Therapy

Adoptive cell therapy (ACT), also known as cellular immunotherapy, is a form of treatment that uses immune cells to target and potentially eliminate cancer cells. ACT approaches range from isolating a patient’s immune cells (e.g., T lymphocytes, DCs, or NK cells), expanding their number in vitro, and then reinfusing them, to genetically engineered (via gene therapy) autologous immune cells such as chimeric antigen receptor T (CAR-T) cells. In the former case, tumor-infiltrating immune cells are obtained from the enzymatically digested tumor specimens and activated ex vivo; then specific clones, presenting with high avidity for tumor antigens, are selected and expanded before reinfusion. In the latter, patients’ T cells are equipped with engineered synthetic receptors, known as CARs, to recognize and eliminate tumor cells expressing a specific target antigen; importantly, this binding is independent from the MHC receptor, thus possibly overcoming the limitation of the previously described form of ACT. Here, we will focus on treatment strategies based on CAR-T cells, the most investigated and clinically meaningful ACT application of autologous T cells; a schematic representation of the generation of CAR-T cells is depicted in Figure 2c. CARs are composed of four main components: an extracellular target antigen-binding domain, a hinge region, a transmembrane domain, and one or more intracellular signaling domains. Indeed, different subsequent generations of CARs, based on the quality and quantity of co-stimulatory intracellular receptors with tyrosin-kinase activity, in series with the CD3ζ have been designed and proposed for clinical testing. After the success of CAR-T cell-based approaches in hematological malignancies and the approval for clinical use in several countries of CAR-T therapy for the treatment of relapsed or refractory lymphomas, leukemia, or multiple myeloma [107,108,109,110], this specific ACT application has been under investigation in many different solid tumors. Nevertheless, specific issues characterizing solid tumors in contrast to hematological malignancies, such as inefficient trafficking and infiltration at the tumor site, immunosuppressive TME, and antigen heterogeneity, have been limiting the clinical applicability of this treatment strategy [111,112]. Overexpression of mutant EGFR variant III (EGFRvIII), present in about 30% of nGBM [113], has been recognized as a tumor-specific, oncogenic epitope able to mediate increased cellular growth, invasion and angiogenesis, and resistance to chemoradiation in GBM cells [114]. Thus, EGFRvIII has been selected as a putative target and is actually the most explored in clinical trials for CAR-T cells against GBM cells. Nevertheless, taking into account the three completed phase I clinical trials testing EGFRvIII CAR-T cells in GBM (NCT02209376, NCT02664363, and NCT01454596), no clear radiological responses were observed. It is worth noting that in the study by O’Rourke and colleagues, 1 patient had residual stable disease for over 18 months after a single dose of CAR-T EGFRvIII cells; translational analyses on tissue specimens after surgery (available for 7 of the enrolled patients) showed antigen decrease at the site of active GBM in 5 out of 7 patients and increased expression of inhibitory molecules and Tregs, highlighting how important it would be for future studies to understand how to overcome immune changes in the TME [115]. Another tumor-specific antigen, Interleukin 13 receptor α2 (IL13Rα2), linked to GBM poor prognosis, has been used to engineer IL13Ra2-specific CAR-T cells to be infused via a catheter/reservoir system into the resection cavity in a pilot safety and feasibility trial that demonstrated fair tolerability and transient anti-glioma responses in 2 out of 3 patients treated [116]. Thus, a phase I trial has been designed with modified co-stimulation and the linker of the IL13Ra2-specific CAR-T to improve antitumor efficacy [117]. Multiple infusions of CAR-T cells were administered both directly into the tumor cavity and into the ventricular system; no grade 3 or higher TRAEs were observed, and a stable regression of tumor localizations (both intracranial and spinal) lasted for 7.5 months after the first CAR-T infusion [117]. Given that overexpression of human epidermal growth factor receptor 2 (HER2) is well known to drive carcinogenesis in several different solid tumors (such as breast cancer, gastric cancer, CRC, and NSCLC) and that it is also found in around 80% of GBMs [118], HER2 CAR-T cell therapy has been extensively explored, with mixed clinical results and concerns about toxicities due to off-target effects [119]. In a phase I trial of HER2-positive GBM testing HER2-CAR CMV pp65-bispecific cytotoxic T lymphocytes, no DLTs were observed, 1 patient experienced PR for more than 9 months, and 7 patients had SD lasting 8 weeks to 29 months, with a mOS of 11.1 months (95% CI, 4.1–27.2 months), leaving an open window for further exploring this therapeutic target [120]. In Table 4, clinical trials testing CAR-T cell therapy in GBM are summarized. Due to the specificities of GBM immunosuppressive TME and the peculiarities of manufacturing CAR-T cells able to persist at the tumor site without off-target side effects and eluding immune-mediated exhaustion, strong evidence is still lacking on the efficacy and potential clinical applicability of the CAR-T cell therapeutic strategy.

## 4. How to Overcome Immunoresistance: Future Perspectives

The TME milieu in GBM harbors peculiar immunosuppressive characteristics linked, as discussed above, to the presence of pro-tumorigenic TAMs that are able to favor proliferation, epithelial-mesenchymal transformation, angiogenesis, and to attract immunosuppressive cells such as MDSCs and Tregs [18,19,123]. These immunosuppressive CD4+CD25+FoxP3+ Tregs are increased in GBM and suppress CD8+ T cell effector functions [51,52]. Also, TANs are capable of promoting tumor cell proliferation, angiogenesis, and tumor necrosis, a correlate of tumor aggressiveness and poor prognosis [23]. In this GBM immune context, T cells have been shown to be very poorly represented (around 5% of CD45+ cells) [39], displaying an exhausted phenotype and functional characteristics compatible with a poorly inflamed and low cytokine-enriched milieu, especially in IDH-wt tumors [5,40]. Thus, it is not surprising that ICB strategies, mostly targeting CD8+CD39+ TILs, did not prove effective in GBM clinical trials, neither in the rGBM [42,63] nor in the nGBM setting [64,124]. The path to be followed in advancing clinical research on immunotherapy for GBM treatment relies on combinatorial treatment strategies, aiming to restore an active immune anti-tumor response and, at the same time, tackle immunosuppression on the TME. Many clinical trials are testing, in nGBM, a strategy combining standard chemoradiation with ICB, either as a combination of anti-PD-1 plus anti-CTLA4 [NCT04396860] or anti-CTLA4 and anti-PD-1/PD-L1 single-agents [ISRCTN84434175, NCT03899857, and NCT03047473]. In some approaches, together with chemotherapy and/or RT with ICB, another immunomodulating agent is being tested, such as an inhibitor of histone deacetylase [NCT03426891], in order to strongly sensitize the TME.

Also, vaccination strategies, which have been shown to interfere with the TME, counteracting some immunosuppressive pathways [105], are being tested in combination with standard chemotherapy and/or RT with or without an ICB, especially in the setting of rGBM, when some degree of chemoradiation resistance is thought to be intrinsic due to previous exposure (NCT04013672, NCT04201873). Moreover, the selection of timing (e.g., neoadjuvant versus adjuvant setting) for clinical testing of immunotherapeutic strategies seems crucial, knowing that both systemic (e.g., chemotherapy) and local (e.g., RT) treatments are able to skew the phenotype of different immune cells in TME [125,126]. In addition, aiming to gain a deeper understanding of the phenotypic and functional dynamic changes of tumor-infiltrating immune cells, surgery after having undergone immunomodulating treatments gives the possibility of tissue collection for analyses. In this direction, some early-phase clinical trials are testing in the neoadjuvant setting (before surgical resection) ICB alone or in combination with other immunomodulating agents (NCT04888611, NCT05465954). Having access to GBM tissue before and eventually after immune treatment, thanks to the combination of single-cell, mass, and spatial localization technologies, will allow us to uncover new mechanisms of innate and acquired immunosuppression and immunotolerance. In a recent study by Liu and colleagues, IL-8 derived from tumor, myeloid, and T cells has been shown to play a pivotal role in orchestrating immunosuppressive pathways and reducing ICB efficacy in GBM, deciphering the potential role of IL-8-CXCR1/CXCR2 axis blockade in restoring the anti-PD-1 effect [127].

## 5. Conclusions

The CNS has long been considered an immunoprivileged site due to the presence of the BBB and the peculiar immunosuppressive characteristics of the parenchymal immune cells, in order to avoid neuro-inflammation [128]. Thus, tumors arising in the brain, both primary and metastatic, have been treated as “cold” tumors, not susceptible to immunomodulatory treatments. This paradigm has shifted with the studies ex vivo, at single-cell level, of brain tumors; metastatic lesions have been proven to harbor immune characteristics that remind the organ of origin [5,40,129], and in some cases, for immunosensitive primary tumors such as melanoma and NSCLC, strong clinical responses with nearly equal intracranial and extracranial disease control rates have been demonstrated with the use of ICB [130,131,132]. Nevertheless, primary brain tumors, and specifically the most aggressive and lethal GBM, have been shown to retain immune hallmarks of strong immunosuppressive TME, and no single immunomodulating strategy of treatment has proved to be clinically effective beyond the standard chemoradiation treatment [42,63,64]. Nonetheless, the door for immunotherapy in GBM is not permanently closed; with new advanced technologies, we are keeping on gaining more and more insight into disease-specific immune features, and combinatorial treatment strategies are being designed and tested to overcome immune resistance. Moreover, phase 0 and window opportunity trials have demonstrated to be very useful for studying the mechanism of action of immunotherapeutic agents in modifying and shaping the GBMs TME, thus in the era of precision medicine. We are currently living in oncology; this would be the best path to pursue to test new immunomodulatory agents.

## Figures and Tables

**Figure 1 cancers-16-00603-f001:**
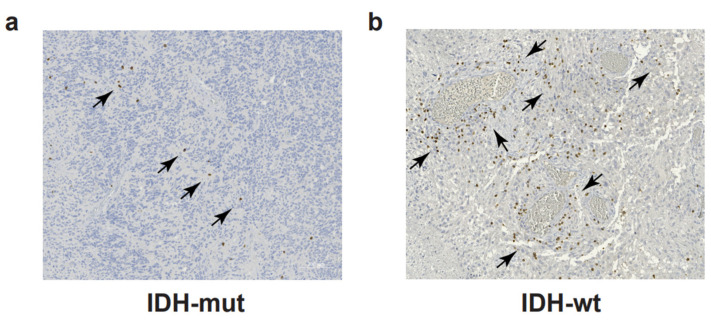
T cell infiltration in glioblastoma. Immunohistochemistry to detect CD3 and DAB chromogen in a representative section of IDH-mut (**a**) and IDH-wt (**b**) glioblastoma samples. Black arrows indicate clusters of CD3+ T cells (in brown). Scale bars: 100 μm.

**Figure 2 cancers-16-00603-f002:**
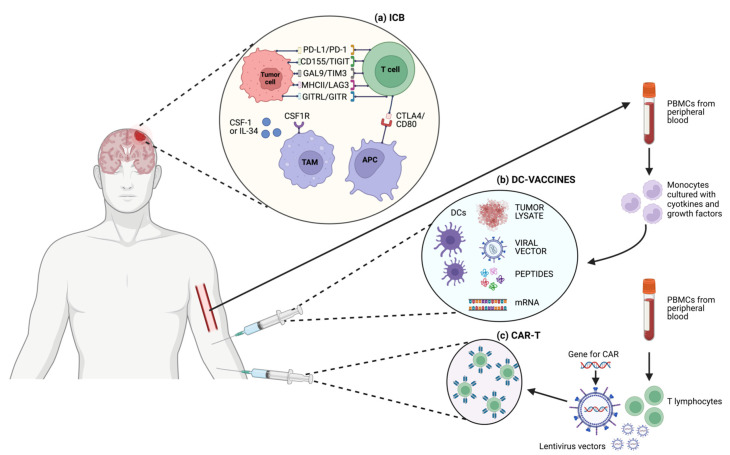
Schematic representation of different T cell-based immunotherapeutic strategies and targets in GBM. (**a**) ICB; (**b**) DC-vaccines; (**c**) CAR-T cells.

**Table 1 cancers-16-00603-t001:** Clinical trials (with published results) exploring the use of anti-PD-1/PD-L1 therapy in GBM.

Title	Identifier	Study Design	Treatment	Population	Main Results	Main TRAEs
A Randomized Phase 3 Open Label Study of Nivolumab vs. Temozolomide in Combination with Radiation Therapy in Newly Diagnosed Adult Subjects with Unmethylated MGMT Glioblastoma (CheckMate 498) [64]	NCT02617589	Phase 3	A: nivolumab + radiotherapy B: temozolomide + radiotherapy	Newly diagnosed methylated GBM	A: 13.0 months B: 14.2 months	A: 21.9% G3-4 TRAEs B: 25.1% G3-4 TRAEs
Phase III trial of chemoradiotherapy with temozolomide plus nivolumab or placebo for newly diagnosed glioblastoma with methylated MGMT promoter (CheckMate548) [65]	NCT02667587	Phase 3	A: nivolumab + temozolomide + radiotherapy B: placebo + temozolomide + radiotherapy	Newly diagnosed methylated GBM	A: mPFS 10.6 months; mOS 31.3 months B: mPFS 10.3 months; mOS 33 months	A: 52.4% G3-4 TRAEs B: 33.6% G3-4 TRAEs
Effect of Nivolumab vs. Bevacizumab in Patients with Recurrent Glioblastoma: The CheckMate 143 Phase 3 Randomized Clinical Trial [63]	NCT02017717	Phase 3	A: nivolumab B: bevacizumab	Recurrent GBM	A: mOS: 9.8 months B: mOS 10 months	A: 18.1% G3-4 TRAEs B: 15.2% G3-4 TRAEs
Circulating Immune Cells and Outcome Analysis from the Phase II Study of PD-L1 Blockade with Durvalumab for Newly Diagnosed and Recurrent Glioblastoma [68]	NCT02336165	Phase 2	A: RT + durvalumab B: durvalumab B2: durvalumab + bevacizumab B3: bevacizumab C: durvalumab + bevacizumab	Newly diagnosed unmethylated GBM Recurrent GBM	A: mOS 15.1 months B: mOS 6.7 months B2: mOS 8.7 months B3: mOS 9.3 months C: mOS 4.5 months	Fatigue G>2: 18% Hyperlypase G>2: 8.8% Hyperamilase G>2: 5.6% Hypertension G>2: 5.6%
Tremelimumab and Durvalumab in Combination or Alone in Treating Patients with Recurrent Malignant Glioma	NCT02794883	Phase 2	A: tremelimumab B: durvalumab C: tremelimumab + durvalumab	Recurrent GBM	A: mOS 7.2 months B: mOS 11.7 months C: mOS 7.7 months	NR
Randomized Phase II and Biomarker Study of Pembrolizumab plus Bevacizumab versus Pembrolizumab Alone for Patients with Recurrent Glioblastoma [69]	NCT02337491	Phase 2	A: pembrolizumab + bevacizumab B: pembrolizumab + placebo	Recurrent GBM	A: mOS 8.8 months B: mOS 10.3 months	A: 20% Hypertension G3 B: 10% Headache G3
Nivolumab with DC Vaccines for Recurrent Brain Tumors	NCT02529072	Phase 1	A: pre-surgery nivolumab B: pre-surgery nivolumab + DC vaccine	Grade III and Grade IV brain tumors	A: mOS 8 months B: mOS 15.3 months	NR
Study of Cabiralizumab in Combination with Nivolumab in Patients With Selected Advanced Cancers	NCT02526017	Phase 1	cabiralizumab + nivolumab	GBM Other advanced solid tumors	mOS 8 months	NR
Intracerebral administration of CTLA-4 and PD-1 immune checkpoint-blocking monoclonal antibodies in patients with recurrent glioblastoma: a phase I clinical trial [70]	NCT03233152	Phase 1	nivolumab (iv) + ipilimumab (intratumoral)	Recurrent GBM	mOS 37 weeks mPFS 11.7 weeks	NR
Combined immunotherapy with controlled interleukin-12 gene therapy and immune checkpoint blockade in recurrent glioblastoma: An open-label, multi-institutional phase I trial [71]	NCT04006119	Phase 1	Cemiplimab-rwlc + Ad-RTS-hIL-12 (intratumoral) + Veledimex (oral)	Recurrent GBM	mOS 8.9 months	NR

GBM: glioblastoma; G: grade; TRAEs: treatment-related adverse events; mOS: median overall survival; mPFS: median progression-free survival; RT: radiotherapy; NR: not reported.

**Table 2 cancers-16-00603-t002:** Ongoing clinical trials exploring combinations of ICB, anti-angiogenic agents, or integration with other innovative immunotherapeutic strategies.

Title	Identifier	Study Design	Treatment	Population	Recruitment Status
Neoadjuvant PD-1 Antibody Alone or Combined with Autologous Glioblastoma Stem-like Cell Antigen-Primed DC Vaccines (GSC-DCV) for Patients With Recurrent Glioblastoma: A Phase II, Randomized Controlled, Double Blind Clinical Trial	NCT04888611	Phase 2	A: camrelizumab + GSC-DCV B: camrelizumab + placebo	Recurrent GBM	Recruiting
Testing the Use of the Immunotherapy Drugs Ipilimumab and Nivolumab Plus Radiation Therapy Compared to the Usual Treatment (Temozolomide and Radiation Therapy) for Newly Diagnosed MGMT Unmethylated Glioblastoma	NCT04396860	Phase 2/3	A: TMZ+RT B: TMZ+RT + ipilimumab + nivolumab	Newly diagnosed unmethylated GBM Gliosarcoma	Active, not recruiting
Efficacy and Safety Study of Neoadjuvant Efineptakin Alfa (NT-I7) and Pembrolizumab in Recurrent Glioblastoma [72]	NCT05465954	Phase 2	Efineptakin Alfa + pembrolizumab	Recurrent GBM	Recruiting
A phase II open-label, randomized study of ipilimumab with temozolomide versus temozolomide alone after surgery and chemoradiotherapy in patients with recently diagnosed glioblastoma: the Ipi-Glio trial protocol [73]	ISRCTN84434175	Phase 2	A: temozolomide + ipilimumab B: temozolomide	Newly diagnosed GBM	Active, not recruiting
A Phase I/II Multicenter Trial Evaluating the Association of Hypofractionated Stereotactic Radiation Therapy and the Anti-Programmed Death-ligand 1 (PD-L1) Durvalumab (Medi4736) for Patients With Recurrent Glioblastoma (STERIMGLI) [74]	NCT02866747	Phase 1/2	A: hFSRT B: durvalumab + hFSRT	Recurrent GBM	Active, not recruiting
Phase Ib/II Trial of Anti-PD-1 Immunotherapy and Stereotactic Radiation in Patients with Recurrent Glioblastoma	NCT04977375	Phase 1/2	pembrolizumab + stereotactic RT	Recurrent GBM	Recruiting
Pembrolizumab and Vorinostat Combined With Temozolomide for Newly Diagnosed Glioblastoma	NCT03426891	Phase 1	TMZ+RT + pembrolizumab + vorinostat	Newly diagnosed GBM	Active, not recruiting
A Phase II Study of Checkpoint Blockade Immunotherapy in Patients with Somatically Hypermutated Recurrent WHO Grade 4 Glioma	NCT04145115	Phase 2	nivolumab + ipilimumab	Recurrent Somatically Hypermutated GBM	Recruiting
Phase II Study of Pembrolizumab Plus SurVaxM for Glioblastoma at First Recurrence [75]	NCT04013672	Phase 2	A: SurVaxM + pembrolizumab (no prior anti-PD1 therapy) B: SurVaxM + pembrolizumab (prior anti-PD1 therapy)	Recurrent GBM	Active, not recruiting
Phase II Study of Pembrolizumab (MK-3475) in Combination with Standard Therapy for Newly Diagnosed Glioblastoma	NCT03197506	Phase 2	A: neoadj and adj pembrolizumab (TMZ+RT) B: adj pembrolizumab (TMZ+RT)	Newly diagnosed GBM	Recruiting
Pembrolizumab for Newly Diagnosed Glioblastoma (PERGOLA)	NCT03899857	Phase 2	TMZ+RT + pembrolizumab	Newly diagnosed GBM	Active, not recruiting
Avelumab in Patients With Newly Diagnosed Glioblastoma Multiforme (SEJ) [76]	NCT03047473	Phase 2	TMZ+RT + avelumab	Newly diagnosed GBM	Completed
Phase I/II Study to Evaluate the Safety and Clinical Efficacy of Atezolizumab (Anti-PD-L1) in Combination With Cabozantinib in Patients with Recurrent Glioblastoma (rGBM)	NCT05039281	Phase 1/2	cabozantinib + atezolizumab	Recurrent GBM	Recruiting
Combination of Checkpoint Inhibition and IDO1 Inhibition Together With Standard Radiotherapy or Chemoradiotherapy in Newly Diagnosed Glioblastoma. A Phase 1 Clinical and Translational Trial	NCT04047706	Phase 1	A (methylated): BMS-986205 + nivolumab (TMZ+RT) B (unmethylated): BMS-986205 + nivolumab (RT)	Newly diagnosed GBM	Active, not recruiting
ASP8374 + Cemiplimab in Recurrent Glioma	NCT04826393	Phase 1	ASP8374 + cemiplimab	Recurrent GBM	Active, not recruiting
Pembrolizumab and a Vaccine (ATL-DC) for the Treatment of Surgically Accessible Recurrent Glioblastoma	NCT04201873	Phase 1	A: DC Tumor Cell Lysate Vaccine + pembrolizumab B: DC Tumor Cell Lysate Vaccine + placebo	Recurrent GBM	Recruiting
Pharmacodynamic Study of Pembrolizumab in Patients With Recurrent Glioblastoma	NCT02337686	Phase 2	Pembrolizumab	Recurrent GBM	Active, not recruiting
An Open-Label, Multi-Center Trial of INO-5401 and INO-9012 Delivered by Electroporation (EP) in Combination With REGN2810 in Subjects With Newly-Diagnosed Glioblastoma (GBM)	NCT03491683	Phase 1\2	A (unmethylated): INO-5401 + INO-9012 + cemiplimab (TMZ+RT) B (methylated): INO-5401 + INO-9012 + cemiplimab + nivolumab (RT)	Newly Diagnosed GBM	Active, not recruiting
A Phase 1 Study to Evaluate IL13Rα2-Targeted Chimeric Antigen Receptor (CAR) T Cells Combined with Checkpoint Inhibition for Patients With Resectable Recurrent Glioblastoma	NCT04003649	Phase 1	A: IL13Rα2-CAR-T + ipilimumab + nivolumab B: IL13Rα2-CAR-T + nivolumab C: IL13Rα2-CAR-T	Recurrent GBM	Recruiting
Phase I Study of PD-L1 Inhibition With Avelumab and Laser Interstitial Thermal Therapy (LITT) in Patients with Recurrent Glioblastoma [77]	NCT03341806	Phase 1	A: avelumab B: avelumab + MRI-guided LITT	Recurrent GBM	Completed

GBM: glioblastoma; RT: radiotherapy; TMZ: temozolomide.

**Table 3 cancers-16-00603-t003:** Clinical trials (with published results) testing DC vaccination in GBM.

Title	Identifier	Study Design	Vaccine Production	Population	Main Results	Treatment Related Adverse Events (TRAEs)
A Phase III Clinical Trial Evaluating DCVax^®^-L, Autologous Dendritic Cells Pulsed With Tumor Lysate Antigen For The Treatment Of Glioblastoma Multiforme (GBM) [96]	NCT00045968	Phase 3	Tumor lysate	Newly diagnosed GBM Recurrent GBM	nGBM mOS: 19.3 months rGBM mOS: 13.2 months	G 3\4 TRAEs: 2 cases of intracranial edema (G3), 1 case of nausea (G3), and 1 case of lymph node infection (G3)
Phase II Trial of Autologous Dendritic Cells Loaded With Autologous Tumor Associated Antigens (AV-GBM-1) as an Adjunctive Therapy Following Primary Surgery Plus Concurrent Chemoradiation in Patients With Newly Diagnosed Glioblastoma [98]	NCT03400917	Phase 2	Lysate of irradiated autologous tumor-initiating cells	Newly diagnosed GBM	mOS: 16 months 2-years OS: 27% mPFS: 10.4 months	G 3\4 TRAEs: none
A Randomized, Double-blind, Controlled Phase IIb Study of the Safety and Efficacy of ICT-107 in Newly Diagnosed Patients With Glioblastoma Multiforme (GBM) Following Resection and Chemoradiation [99]	NCT01280552	Phase 2	Synthetic peptide epitopes targeting GBM associated antigens MAGE-1, HER-2, AIM-2, TRP-2, gp100, and IL13Rα2	Newly diagnosed GBM	mOS: 17 months (vs. 15 months, NS) mPFS: 11.2 months (vs. 9 months, *p* = 0.011)	G 3\4 TRAEs: no difference from the control group (G3 nervous system disorders, fatigue, WBC decrease, and infections)
A Phase II Feasibility Study of Adjuvant Intra-Nodal Autologous Dendritic Cell Vaccination for Newly Diagnosed Glioblastoma Multiforme	NCT00323115	Phase 2	Tumor lysate	Newly diagnosed GBM	mOS: 28 (15-44) months mPFS: 9.5 (5-41) months	N.A.
Phase I Trial of Vaccination With Autologous Dendritic Cells Pulsed With Lysate Derived From an Allogeneic Glioblastoma Stem-like Cell Line for Patients With Newly Diagnosed or Recurrent Glioblastoma [100]	NCT02010606	Phase 1	Lysate from an allogeneic stem-like cell line	Newly diagnosed GBM Recurrent GBM	nGBM mOS: 20.36 months mPFS: 8.75 months rGBM mOS: 11.97 months mPFS: 3.23 months	G 3\4 TRAEs: none
A phase I trial of surgical resection with Gliadel Wafer placement followed by vaccination with dendritic cells pulsed with tumor lysate for patients with malignant glioma [101]	N.A.	Phase 1	Tumor lysate	Newly diagnosed GBM Recurrent GBM	nGBM mOS: 27.7 months mPFS: 4.8 months rGBM mOS: 10.9 months mPFS: 1.9 months	G 3\4 TRAEs: none
Anti-Tumor Immunotherapy Targeted Against Cytomegalovirus in Patients With Newly-Diagnosed Glioblastoma Multiforme During Recovery From Therapeutic Temozolomide-induced Lymphopenia	NCT00639639	Phase 1	CMV pp65	Newly diagnosed GBM	ATTAC-GM mOS: 37.7 months ATTAC-Td mOS: 38.3 months	N.A.
Evaluation of Overcoming Limited Migration and Enhancing Cytomegalovirus-specific Dendritic Cell Vaccines With Adjuvant TEtanus Pre-conditioning in Patients With Newly-diagnosed Glioblastoma	NCT02366728	Phase 2	CMV pp65	Newly diagnosed GBM	Arm1 (unpulsed DC): mOS 16 months Arm2 (Td pre-conditioning): mOS 20 months Arm3 (Basiliximab + Td pre-conditioning): mOS 19 months	SAEs: Arm 1: 1\27 pts (lung infection) Arm 2: 4\28 pts (urinary infection, nervous system, and psychiatric disorders) Arm 3: 2\9 pts (colon perforation, nervous system disorders)

nGBM: newly diagnosed glioblastoma; rGBM: recurrent glioblastoma; G: grade; TRAEs: treatment-related adverse events; mOS: median overall survival; mPFS: median progression-free survival; N.A.: not applicable; SAEs: severe adverse events.

**Table 4 cancers-16-00603-t004:** Clinical trials (ongoing and with results) exploring CAR-T cell therapy in GBM.

Title	Identifier	Study Design	Target Antigen	Status	Main Results	Main TRAEs
Administration of HER2 Chimeric Antigen Receptor Expressing CMV-Specific Cytotoxic T Cells In Patients With Glioblastoma Multiforme (HERT-GBM)	NCT01109095	Phase 1	HER2	Completed	16 evaluable pts (9 adults and 7 children) 1 PR (>9 months); 7 SD (8 weeks to 29 months); 8 PD mOS 11.1 months (95% CI, 4.1–27.2 months)	No DLTs G2 seizures and/or headaches
Pilot Study of Autologous T Cells Redirected to EGFRVIII with a Chimeric Antigen Receptor in Patients With EGFRVIII+ Glioblastoma [121]	NCT02209376	Phase 1	EGFRvIII	Terminated	mOS: 251 days (~8 months) (1 patient alive at 18 months) mPFS: not evaluable	No dose-limiting toxicities AEs: neurological events (seizures and neurologic decline), fatigue, fever, muscle weakness, skin and subcutaneous tissue disorders
EGFRvIII Chimeric Antigen Receptor (CAR) Gene-Modified T Cells for Patients with Newly-Diagnosed GBM During Lymphopenia [115]	NCT02664363	Phase 1	EGFRvIII	Terminated	This study terminated before a MTD could be determined	SAEs: Generalized muscle weakness and confusion Other AEs: Anemia, nausea, vomiting, WBC and PLT count decrease, hyperglycemia, hypocalcemia, cognitive disturbance, hypertension
A Phase I/II Study of the Safety and Feasibility of Administering T Cells Expressing Anti-EGFRvIII Chimeric Antigen Receptor to Patients With Malignant Gliomas Expressing EGFRvIII [122]	NCT01454596	Phase 1	EGFRvIII	Completed	No clinical responses were observed The protocol was closed rather than proceed into the Ph II portion	SAEs: 1 death due to multi-organ failure and G3/4 dyspnea Other TRAEs: febrile neutropenia, arrhythmia, diarrhea, WBC and platelet count decrease, transaminitis, creatinine increase, electrolyte disorders, confusion/dizziness, thrombosis
Phase 1 Study of EGFRvIII-Directed CAR T Cells Combined with PD-1 Inhibition in Patients With Newly Diagnosed: MGMT-Unmethylated Glioblastoma	NCT03726515	Phase 1	EGFRvIII	Completed	No results posted	No results posted
Phase I Study of Cellular ImmunoTx Using Memory Enriched T Cells Lentivirally Transduced to Express an IL13Rα2-Specific, Hinge-Optimized, 41BB-Costimulatory Chimeric Receptor and a Truncated CD19 for Pts With Rec/Ref Malignant Glioma	NCT02208362	Phase 1	IL13Rα2	Active, not recruiting	--	--
A Phase I Clinical Study to Evaluate the Safety, Tolerability, Pharmacokinetics, and Antitumor Activity of SNC-109 CAR-T Cell Therapy in Subjects With Recurrent Glioblastoma	NCT05868083	Phase 1	HER2, IL13Rα2, EGFR, EGFRvIII	Recruiting	--	--
A Pilot Study of Chimeric Antigen Receptor (CAR) T Cells Targeting B7-H3 Antigen in Treating Patients with Recurrent and Refractory Glioblastoma	NCT04385173	Phase 1	B7-H3	Recruiting	--	--
B7-H3-Targeted Chimeric Antigen Receptor (CAR) T Cells in Treating Patients With Recurrent or Refractory Glioblastoma	NCT04077866	Phase 1	B7-H3	Recruiting	--	--
An Open, Single-Arm Phase 1 Study to Evaluate the Safety/Preliminary Effectiveness and Determine the Maximal Tolerated Dose of B7-H3-targeting CAR-T Cell Therapy in Treating Recurrent Glioblastomas	NCT05241392	Phase 1	B7-H3	Recruiting	--	--
A Phase 1 Study to Evaluate IL13Rα2-Targeted Chimeric Antigen Receptor (CAR) T Cells Combined with Checkpoint Inhibition for Patients With Resectable Recurrent Glioblastoma	NCT04003649	Phase 1	IL13Rα2	Recruiting	--	--
A Phase I Clinical Trial of NKG2D-based CAR T-cell Injection for Subjects With Relapsed/Refractory NKG2DL+ Solid Tumors	NCT05131763	Phase 1	NKG2D	Recruiting	--	--
A Phase 1 Study to Evaluate Chimeric Antigen Receptor (CAR) T Cells with a Chlorotoxin Tumor-Targeting Domain for Patients with MMP2+ Recurrent or Progressive Glioblastoma	NCT04214392	Phase 1	Chlorotoxin	Recruiting	--	--
A Phase 1 Study to Evaluate IL13Rα2-Targeted Chimeric Antigen Receptor (CAR) T Cells for Adult Patients with Leptomeningeal Glioblastoma, Ependymoma, or Medulloblastoma	NCT04661384	Phase 1	IL13Rα2	Recruiting	--	--
Loc3CAR: Locoregional Delivery of B7-H3-specific Chimeric Antigen Receptor Autologous T Cells for Pediatric Patients With Primary CNS Tumors	NCT05835687	Phase 1	B7-H3	Recruiting	--	--
INCIPIENT: Intraventricular CARv3-TEAM-E T Cells for Patients With GBM	NCT05660369	Phase 1	EGFRvIII	Recruiting	--	--
Phase I Study of Intraventricular Infusion of T Cells Expressing B7-H3 Specific Chimeric Antigen Receptors (CAR) in Subjects With Recurrent or Refractory Glioblastoma	NCT05366179	Phase 1	B7-H3	Recruiting	--	--
Phase I Clinical Trial of Locoregionally (LR) Delivered Autologous B7-H3 Chimeric Antigen Receptor T-Cells (B7-H3CART) in Adults with Recurrent Glioblastoma Multiforme	NCT05474378	Phase 1	B7-H3	Recruiting	--	--
Phase 1, Open-label Study Evaluating the Safety and Feasibility of CART-EGFR-IL13Ra2 Cells in Patients with EGFR-Amplified Recurrent	NCT05168423	Phase 1	IL13Rα2, EGFR	Recruiting	--	--
Phase I Study of Cellular Immunotherapy Using Memory-Enriched T Cells Lentivirally Transduced to Express a HER2-Specific, Hinge-Optimized, 41BB-Costimulatory Chimeric Receptor and a Truncated CD19 for Patients with Recurrent/Refractory Malignant Glioma	NCT03389230	Phase 1	HER2	Active, not recruiting	--	--
Phase 1 Study of Autologous Tris-CAR-T Cell Locoregional Immunotherapy for Recurrent Glioblastoma	NCT05577091	Phase 1	IL7Rα	Not yet recruiting	--	--
Pilot Study of UWNKG2D CAR-T in Treating Patients with Recurrent Glioblastoma	NCT04717999	Phase 1	NKG2D	Not yet recruiting	--	--
An Open Clinical Study to Evaluate the Safety, Tolerance, and Initial Efficacy of Epidermal Growth Factor Receptor Variant III Chimeric Antigen Receptor T (EGFRvIII CAR-T) in the Treatment of Recurrent Glioblastoma	NCT05802693	Phase 1	EGFRvIII	Not yet recruiting	--	--

Pts: patients; PR: partial response; PD: progressive disease; SD: stable disease; DLTs: dose-limiting toxicities; G: grade; mOS: median overall survival; mPFS: median progression-free survival; SAEs: adverse events; WBC: white blood cells; PLT: platelets; MTD: maximum tolerated dose.

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
