# Peer review of "T Cell Features in Glioblastoma May Guide Therapeutic Strategies to Overcome Microenvironment Immunosuppression"

_cancers, 2024, doi:10.3390/cancers16030603_

Round 1

Reviewer 1 Report

Comments and Suggestions for Authors

Authors review the progress of GBM TME, including miacroglia and mascrophages, neutrophils, DC, and T cells. They also summarize the progress in immunotherapy. It is helpful for glioma researchers.

Comments:

1.       Figure 1 is not good. It may be removed.

2.       Microglia and macrophages, together accounting for the majority (around 30%) of 90 TME cellularity, constitute the innate immunity of the CNS and are responsible for the 91 maintenance of the normal brain function and homeostasis.” The citation is lost.

3.       Some new progress of GBM TME should be dicussed, such as rGBM TME.

Reviewer 2 Report

Comments and Suggestions for Authors

The review by Losurdo and coll. explores the prospective use of T cells in glioblastoma therapy. The topic is interesting, yet I have some suggestions to improve the text.

- please add 'MAY' before 'guide' in the title, since the evidence presented in the paper is not conclusive.

- the resolution of fig. 1 is too low and is impossible to evaluate (no black arrows are visible).

- please define each acronym at the first occurrence (e.g.: OS…), and define all abbreviations used in each table and figure (even if trivial. But ‘ms’ is the usual abbreviation for ‘milliseconds’ which seems not the case in table 1. And so on), so that they can be read as stand-alone, without reference to the text.

- line 263: it is not clear how was the search for clinical trials performed. Where? When? Inclusion criteria? Exclusion criteria? How was the search for articles performed?

- page 24, chapter 4 title: ‘prospective’ is an adjective. Do the Authors mean ‘Perspectives’? 

- there are various English errors which should be amended before publication, as detailed below.

Comments on the Quality of English Language

Language issues:

-       Line 15: ‘diagnosed’, not ‘diagnoses’

-       Line 91: ‘cellularity’ probably means ‘cells’

-       Line 109: ‘at a spatial level’ is complex to understand (I guess it means concerning spatial relationships)

-       Line 173: delete ‘the’ before the number ‘5%’

-       L. 176 delete ‘,’ between ‘similarly’ and ‘to’

-       L. 181 change ‘physiologic’ to ‘physiological’ and ‘restraining’ to ‘reducing’

-       L. 185: ‘clinic’ = ‘clinics’ or ‘clinical use’?

-       P.24 Line 32: delete the ‘,’ between ‘treatments’ and ‘gives’. 

-       P 24, L 37: please omit ‘us’ after ‘allow’…hopefully, the effort of everyone is welcome in this field
